# Multimodal factor evaluation system for organismal transparency by hyperspectral imaging

Takumi T. Shito[1], Kotaro Oka[1,2,3,4], Kohji Hotta[1] *

1 Department of Bioscience and Informatics, Faculty of Science and Technology, Keio University, Yokohama, Japan, 2 Waseda Research Institute for Science and Engineering, Waseda University, Tokyo, Japan, 3 Graduate Institute of Medicine, College of Medicine, Kaohsiung Medical University, Kaohsiung City, Taiwan, 4 School of Frontier Engineering, Kitasato University, Sagamihara, Japan

* khotta@bio.keio.ac.jp

**Data Availability Statement:** All relevant data are within the paper and its Supporting Information files.

**Funding:** This study was supported in part by JSPS KAKENHI (21H00440, 23H04717), Keio

## Abstract

Organismal transparency constitutes a significant concern in whole-body live imaging, yet its underlying structural, genetic, and physiological foundations remain inadequately comprehended. Diverse environmental and physiological factors (multimodal factors) are recognized for their influence on organismal transparency. However, a comprehensive and integrated quantitative evaluation system for biological transparency across a broad spectrum of wavelengths is presently lacking. In this study, we have devised an evaluation system to gauge alterations in organismal transparency induced by multimodal factors, encompassing a wide range of transmittance spanning from 380 to 1000 nm, utilizing hyperspectral microscopy. Through experimentation, we have scrutinized the impact of three environmental variables (temperature, salinity, and pH) and the effect of 11 drugs treatment containing inhibitors targeting physiological processes in the ascidian *Ascidiella aspersa*. This particular species, known for its exceptionally transparent eggs and embryos, serves as an ideal model. We calculated bio-transparency defined as the mean transmittance ratio of visible light within the range of 400–760 nm. Our findings reveal a positive correlation between bio-transparency and temperature, while an inverse relationship is observed with salinity levels. Notably, reduced pH levels and exposure to six drugs have led to significant decreasing in bio-transparency (ranging from 4.2% to 58.6%). Principal component analysis (PCA) on the measured transmittance data classified these factors into distinct groups. This suggest diverse pathways through which opacification occurs across different spectrum regions. The outcome of our quantitative analysis of bio-transparency holds potential applicability to diverse living organisms on multiple scales. This analytical framework also contributes to a holistic comprehension of the mechanisms underlying biological transparency, which is susceptible to many environmental and physiological modalities.

University Research and Education Center for Natural Sciences Budget, and KLL Keio Leading Program to KH. The Research Institute of Marine Invertebrates (IKU2021-02) supported TTS. The Keio University Doctorate Student Grant-in-Aid Program from Ushioda Memorial Fund supported TTS. JSPS KAKENHI Grant Number JP 22J22628 supported TTS. There was no additional external funding received for this study. The funders had no role in study design, data collection and analysis, decision to publish, or preparation of the manuscript.

**Competing interests:** The authors have declared that no competing interests exist.

## Introduction

Numerous animals exhibit organismal transparency, including but not limited to jellyfish, siphonophores, certain crustaceans, pteropods, some squids, salps, and fish larvae. Organismal transparency is generally thought to be a mechanism for evading visual predation [1–3]. Several studies have assessed changes in biological transparency and have underscored the diverse influences on this trait, encompassing environmental variables [4], structural aspects [5, 6], physiological factors [4, 7], genetic expression [8], developmental stage [9], and more.

Intriguingly, several transparent animals, such as gelatinous zooplankton, experience a loss of transparency when moribund or after death, suggesting that active physiological processes maintain this characteristic [3, 10]. Within this context, certain transparent shrimps become opaque due to a light scattering due to heightened perfusion. This alteration can be prompted by tail-flip escape responses and environmental shifts, such as temperature and salinity [4, 7]. Additional examples are found in glassfrogs, which modulate their body transparency through the sequestration of red blood cells. These cells are either stored within their liver or released from circulation, leading to alterations in transparency levels [11].

In the context of this research, two distinct methods have been employed to quantify changes in transparency. One approach involves calculating transmittance through photography, which is adaptable to varying-sized organisms. However, this method is constrained to generating a three channel RGB image. An alternative technique employs spectroradiometers to assess transmittance by gauging emitted light across numerous channels. Regrettably, this approach is generally feasible only for organisms at the centimeter scale and cannot furnish two-dimensional structural insights. Both of these methodologies encounter limitations in their ability to comprehensively measure transparency across diverse wavelengths and scales. Furthermore, There have been no methods to evaluate and integrate the impacts of multimodal factors that cause alterations in organismal transparency.

In this study, we established an evaluation system for organismal transparency alterations in a wide range of transmittance from 380 to 1000 nm. This innovative methodology employs hyperspectral microscopy and principal component analysis (PCA). Our initial focus is examining the impact of both environmental and chemical factors on the transparency of eggs from the ascidian species *Ascidiella aspersa*. These eggs notably transmit approximately 90% of visible light across the visible light wavelength [12]. Our approach encompassed the manipulation of environmental conditions–specifically temperature, salinity, and pH–and the utilization of 11 compounds containing pharmacological inhibitors. Within this evaluation system, we elucidate how these factors influence organismal transparency and discuss mechanisms orchestrating the observed alterations.

## Materials and methods

### Ethical approval

Ethical approval was not required.

### Sampling *A. aspersa* eggs and drug treatment

Adults of *A. aspersa* that were clinging to fixed scallop ropes were collected from locations in Japan: Onagawa (Onagawa Field Center, Tohoku University) in October 2019 and June 2020, Hakodate (Hokkaido Research Organization, Hakodate Fisheries Research Institute) in June and October 2021, and Yoichi (Hokkaido Research Organization, Central Fisheries Research Institute) in October 2021. Eggs were extracted from the adult gonoducts, then dechorionated using a solution containing 0.05% actinase-E and 1% mercaptoacetic acid sodium salt. These

eggs were subsequently placed in plastic Petri dishes filled with seawater. Glass dishes with cover glasses were used to hold seawaster with varying environmental conditions or added drugs. Following 21-hour incubation period, images were captured using a microscope camera (WRAYCAM-SR300, WRAYMER, Osaka, Japan) attached to a stereomicroscope (OLYMPUS, SZX16, Tokyo, Japan).

## Conditions of egg incubation

The eggs were subjected to incubation in seawater at various temperatures (4, 13, 20, 27, and 30˚C), salinity levels (0, 8, 20, 33, 41, and 55 parts per thousand: ppt), and pH values (1.3, 2.3, 3.4, 5.73, 8.06, and 10.7). Natural seawater was utilized, except for investigations involving different salinities, for which artificial seawater (Tomita Pharmaceutical, Japan). pH adjustments were made using 2N HCl and 2N NaOH in conjunction with natural seawater.

## 11 Drugs containing inhibitors targeting physiological functions

In order to investigate factors influencing the transparency of *A. aspersa* eggs, a range of drugs known for their inhibition of physiological processes were employed (see Table 1 for details). The chosen inhibitors encompassed actinomycin D, a transcription inhibitor (Sigma, St. Louis, USA); cycloheximide, a translation inhibitor (Sigma, St. Louis, USA); 2-deoxy-D-glucose, a glycolysis inhibitor (Wako Pure Chemical, Osaka, Japan); ML-7, an inhibitor of myosin light chain kinase (MLCK) targeting the cytoskeleton (Tocris Bioscience, Bristol Avonmouth, UK); and oligomycin A, an ATPase inhibitor (Cayman Chemical, Michigan, USA). Dimethyl sulfoxide (DMSO) was utilized as the solvent for these drugs, while ethanol was introduced as a teratogen during the embryogenesis stage [13].

**Drug description and final concentration in this study and previous research.** We employed NaCl to examine the impact of the chloride ion concentration, serving as a negative control compared to the HCl treatment, while mannitol was utilized to assess osmotic pressure effects. The final concentration of each inhibitor was determined based on the concentrations reported in previous studies involving ascidian embryos and $IC_{50}$ values for ascidians or other species (Table 1). Eggs that exhibited opacity due to damage during the dechorionation process (which is also observed following the micro-injection of transparent eggs of *Phallusia mammilata* [14]) were defined as dead eggs. The primary focus of this study was to establish an evaluation system; thus, the viability of eggs under different conditions was not within the scope of this research. For analysis, the dead eggs were quantified using hyperspectral microscopy through an inverted microscope and a hyperspectral camera (custom model NH-KO, EBA JAPAN, Tokyo, Japan).

## Calculating and evaluating egg transparency

Egg transparency was assessed 21 h after modifying the environment or administering pharmacological treatments. The section of this 21 h observation interval was determined based on the experiments grass shrimp, for which 18 h or 24 h interval after the environmental change was enough to observe the physiological fluctuations of transparency [4]. Spectral data spanning wavelengths from 380 to 1000 nm in 5-nm intervals were acquired using a hyperspectral camera mounted on an inverted microscope (NIKON Eclipse TE 2000-u) equipped with a 10X objective lens. The transmittance of the eggs ($T$) was calculated as the ratio of the transmitted light intensity ($I$) to the incident light intensity ($I_0$).

Considering that differences in transparencies across individual ascidian batches, the relative transmittance ($Tr$) was derived by multiplying transmittance of the specific eggs under examined ($T$) with a normalization factor. This normalization factor was established as the

**Table 1. Drugs and culture conditions tried in this research and previous usages.**

| Group | Drug / culture condition | This study | | Previous study | | |
|---|---|---|---|---|---|---|
| | | Description | Concentration | Concentration | Organism | ref. |
| Protein synthesis | ActinomycinD | Transcription inhibitor | 100 $\mu$g/ml | 40 $\mu$g/ml | ascidian embryo | [30] |
| | Cycloheximide | Translation inhibitor | 1% | 0.10% | ascidian hemocyte | [31] |
| Energy | 2-Deoxy-D-glucose(2-DG,DOG) | Glycolysis inhibitor | 5 mM | 2 mM | mouse sperm | [32] |
| | oligomycinA | ATPase inhibitor | 4 $\mu$M | 1.23$\mu$M | ascidian juvenile | [33] |
| Cytoskelton | ML-7 | MLCK inhibitor | 100 $\mu$M | 100 $\mu$M | ascidian sperm | [34] |
| Solvent | Ethanol | Higher mortality embryos | 5.00% | 2.90% | zebrafish embryo | [13] |
| | DMSO | Organic Solvent | 0.5%, 10% | | | |
| Salt | NaCl | Salt | 50mM | | | |
| Sugar | Mannitol | osmic pressure | 250mM | | | |

ratio between the typical transparency (at 20˚C, salinity of 33 ppt, and pH 8.06 in seawater) of the given batch of eggs ($T_n$) and the transparency of a reference batch of eggs ($Tn_0$), as outlined in Eq (1).

**Transmittance.**

$$\mathbf{T} = \frac{I}{I_0} \qquad T_r = T\frac{T_n}{T_{n0}}$$  (1)

Bio-transparency [12] was calculated as the average of the hundredfold transmittance values within the visible spectrum (400–760 nm). Since bio-transparency does not account for the influence of eggs thickness, we calculated the attenuation coefficient ($\mu$) using the egg thickness ($x$) through the following equations. To account for batch-related effects on eggs size, we computed the relative expansion rate ($e_r$) was diameter of a current egg ($x$) and intact eggs of same batch ($x_0$). Consequently, the relative attenuation coefficients ($\mu_r$) were calculated according to Eq (2).

**Attenuation coefficient.**

$$\boldsymbol{\mu} = -\frac{1}{x} \ln\tau \quad e_r = \frac{x}{x_0} \quad \mu_r = -\frac{1}{er} \ln\tau_r$$  (2)

The Dunnett test was executed using the multcomp package within R version 3.6.1 (http://www.R-project.org). Dimension reduction to 125 bands of transmittance data was carried out via PCA utilizing in the built-in R packages. Heatmap analysis was also conducted using the Ward clustering method within the default R package.

## Results

### The alterations of ascidian egg transparencies by environmental factors

We established controlled environmental conditions for the seawater at 20˚C, 33 ppt salinity, and pH 8.0, previously employed in general ascidian biological experiments [15, 16]. The transparency of *A. aspersa* eggs is depicted in Fig 1A, quantified through hyperspectral imaging, showing significant transmittance across the ultraviolet to infrared spectrum. The hyperspectral imaging data displayed a gradual increase in transparency across the range up to 1000 nm, except for a pronounced decrease centered around 415 nm, as illustrated in Fig 1B. The calculated bio-transparency was 88.0 ± 0.7% (Fig 1B).

Regarding temperature, the maximum recorded bio-transparency was 89.4% at 4˚C, while the minimum reached 80.4% at 27˚C (Fig 1C and S1 Table). A clear inverse relationship

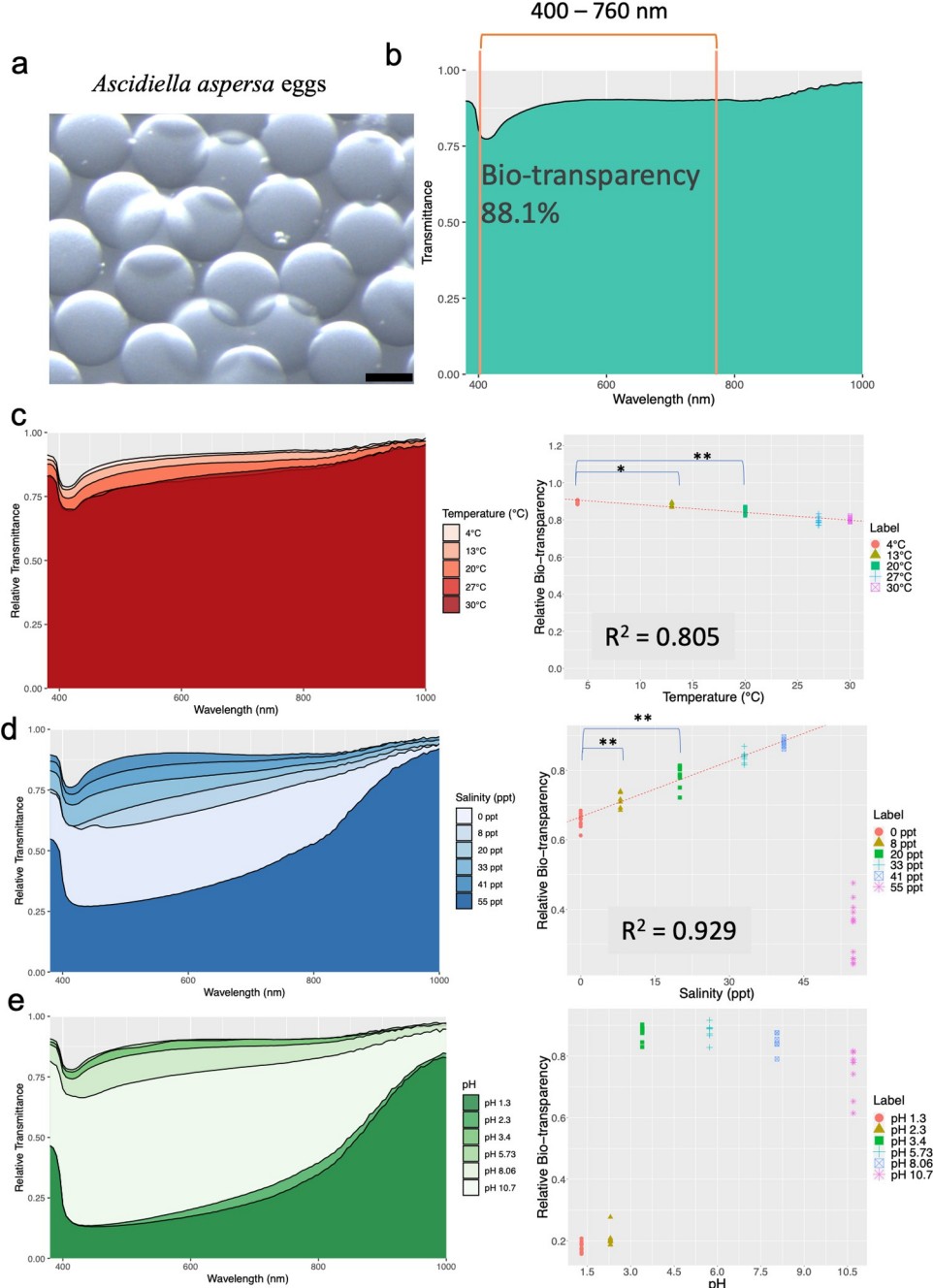

**Fig 1. Egg transparency under various environmental conditions. a** Pictures of *A. aspersa* eggs under a controlled environment (20˚C, 33 ppt, pH 8.06). The scale bar is 100 μm. **b** The transmittance spectrum of a. The average of the visible range (400–760 nm) was calculated as bio-transparency. Egg transmittance spectra and bio-transparencies with linear regression line with changes in **c)** temperature in the range from 4˚C to 30˚C, **d)** salinity from 0 to 41 ppt, and **e)** pH from 1.3 to 10.7. *p<0.1, **p<0.05 with Welch t-test.

between bio-transparency and temperature was observed within the 4–30˚C range, exhibiting a coefficient of determination ($R^2$) value of 0.805. Conversely, salinity displayed a notably strong positive correlation ($R^2$ = 0.929) with bio-transparency. The 60–90% bio-transparency levels corresponded to salinity levels spanning 0–41 ppt (Fig 1D). Notably, the eggs exhibited

signs of rupture at both 0 and 8 ppt salinities. Interestingly, at 55 ppt, the eggs displayed a distinct light brown hue (S1A Fig), which differed from observations under other salinity conditions.

While temperature and salinity alterations are correlated with bio-transparency, no significant correlations were apparent between the bio-transparency and pH ($R^2 = 0.4$). Bio-transparency displayed comparatively minor fluctuations across the pH range of 3.4–8.06, with its pinnacle of 88.1% observed at pH 5.73. Conversely, at pH 10.7, bio-transparency dropped to 74.8%. At the extreme ends of pH spectrum, notably low bio-transparency levels of 18.2% and 20.7% emerged at pH 1.3 and pH 2.3, respectively (Fig 1E). Interestingly, under pH 1.3 and pH 2.3 conditions, the eggs underwent a color transformation to ocher (S1A Fig). There seemed to be an optimal peak value around pH 5.73 with the maximum transparency (Fig 1E).

Considering that transmittance can be influenced by the optical path length, we conducted additional investigations into transparency, accounting for the contribution of egg diameters. This was archieved by calculating the relative expansion rate and deriving relative attenuation coefficients (S2 Table). The observed correlations between transparency and both temperature and salinity were upheld in the context of these attenuation coefficients (S3 Fig). The ranking of relative attenuation coefficients closely mirrored that of bio-transparency, with a few exceptions, such as the ranking of transparency for mannitol and 2DG (S4 Fig). The patterns observed in the heatmap clusters largely remained consistent, with only a few instances of variation, notably the positioning of ML-7 (S3–S6 Figs).

## Alterations of egg transparency by drugs

We extensively investigated alterations in *A. aspersa* egg transparency across the wavelength range of 380–1000 nm, employing treatment with 11 drugs containing pharmacological inhibitors. Relative to the baseline bio-transparency (88.1%) observed under control conditions (20°C, 33 ppt, pH 8.06), the bio-transparency exhibited drastic fluctuations, ranging from 89.4% to 29.5% (yielding differences ranging from +1.2% to -58.6% compared to the control, as detailed in S1 Table). Notably, the application of four out of the 11 drugs induced alterations in the specific transparency spectra of egg transparency (Fig 2A). The transparency spectra were notably transformed through treatment with the MLCK inhibitor ML-7, resulting in a sharp decline centered at 415 nm (Fig 2A, ML-7; S2 Fig). Remarkably, exposure to 10% DMSO led to substantial spectrum changes, culminating tin a minimal bio-transparency of 29.3% at 490 nm. Under stereomicroscopy, these conditions elicited a notable shift in egg appearance from transparent to white (S1B Fig). Bio-transparency values were computed as the average within the 400–760 nm range (Fig 2B). Among the 11 drugs tested, seven drugs exhibited significant differences in bio-transparency (cycloheximide: 83.9 ± 2.2%, representing -4.2% decrease; oligomycin A: 82.3 ± 1.8%, corresponding to a -5.8% change; ML-7: 57.3 ± 4.1%, reflecting a -30.8% change; dead eggs: 36.4 ± 3.5%, indicating a -51.7% change; 10% DMSO: 34.7 ± 3.5%, signifying a -53.4% alteration relative to control conditions (S1 Table).

A concentration of 0.5% DMSO, mirroring the solvent concentration used for other drugs, exhibited negligible impact on bio-transparency (-1.0%). However, an elevated DMSO concentration (10%) notably reduced bio-transparency by -53.4%. Furthermore, the bio-transparency of dead eggs also experienced a significant reduction of -51.7%. In comparison, the effect induced by ML-7 was relatively milder, resulting in a bio-transparency decrease of -30.8%. Among the inhibitors tested, the transcription inhibitor cycloheximide and the ATPase inhibitor oligomycin-A exhibited modest decreases in bio-transparency at -4.2% and -5.8%, respectively. Conversely, the nucleic acid and protein synthesis inhibitor actinomycin-D and the glycolysis inhibitor 2DG caused slight changes with no significant deviations of +0.9% and

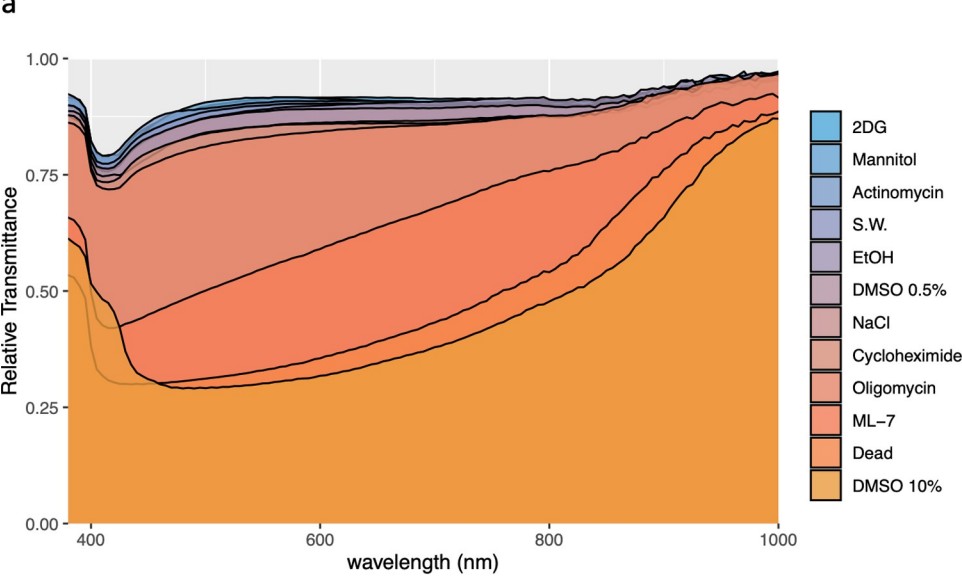

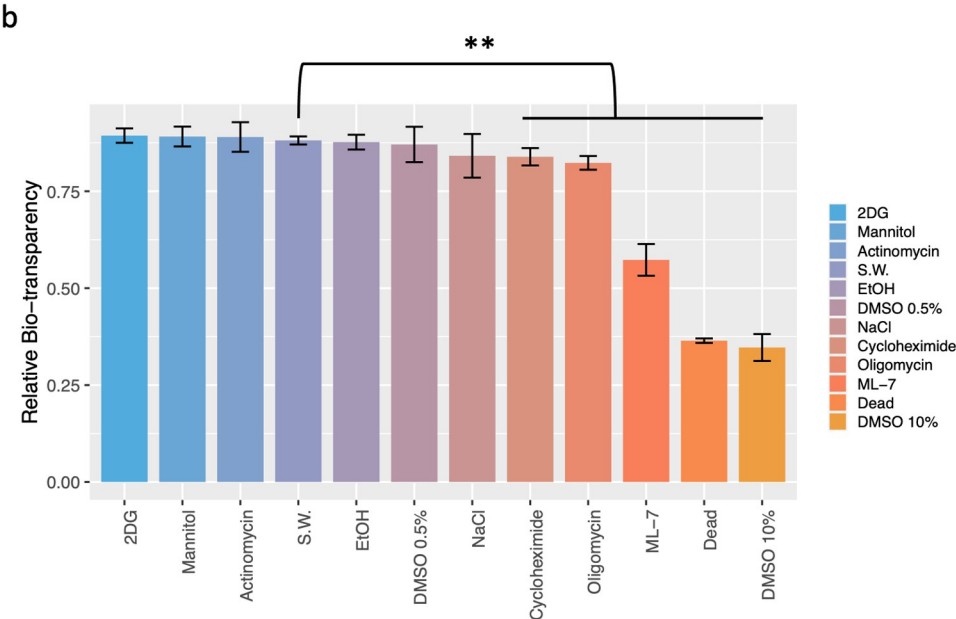

**Fig 2. Alterations of egg transparency by drugs containing inhibitors of physiological functions. a** Transmittance spectra. **b** Comparison of bio-transparency of eggs. Error bars show s.d. **p<0.05 compared with control seawater with Dunnett's test.

+1.2%, respectively. Ethanol, mannitol, and NaCl showed non-significant alterations of -0.4%, +1.2%, and -4.0%, respectively.

## Visualization with transparency roadmap using PCA analysis and clustering

The spectral influence on transparency might differ for each factor, prompting us to attempt their isolation through PCA. Employing PCA facilitated dimensionality reduction by

condensing transmitted light spectrum, comprising 125 bands spanning 380 nm to 1000 nm, into two-dimensional data. Combining transmittance data from both environmental alteration experiments (Fig 1) and drugs experiments (Fig 2), we extracted PC1 and PC2. These components accounted for 98.8% and 0.81% of overall variance, respectively (Fig 3A). PC1 seemed indicative of bio-transparency, as evidenced by the marked decline in bio-transparency corresponding to higher PC1 values (such as observed with pH 1.3 and 2.3, 10% DMSO, salinity of 55 ppt, dead eggs, and ML-7).

While the contribution of PC2 appears relatively modest, it does enable the classification of certain environmental factors and drugs based on their spectral characteristics. For instance, ML-7 exhibited an elevated PC2 value, while 10% DMSO displayed a diminished PC2 value. Employing the Ward method, we conducted hierarchical clustering of the bio-transparency and depicted the results through a heatmap (Fig 3B). Notably, the clustering pattern aligned with groups of impacted wavelengths, effectively dividing them into suggested clades: a composite clade spanning 380–395, 700–875, and 880–1000 nm, as well as distinct clades spanning 400–490 and 495–695 nm.

## Discussion

In this study, we have effectively developed an experimental system for quantitatively evaluating the impact of various modalities on organismal transparency. These modalities encompass environmental factors like temperature, pH, and osmotic pressure and molecular factors such as inhibitors. This evaluation was achieved by integrating hyperspectral imaging with PCA and clustering analysis. Initially, we employed this methodology to examine the transparent eggs of an ascidian, a marine invertebrate. We then categorized the observed effects of environmental factors and drugs containing pharmacological inhibitors on organismal transparency.

### Relationships between temperature and bio-transparency

Temperature exhibited a negative correlation with bio-transparency within the temperature range of 4˚C to 30˚C (Fig 1C, $R^2$ = 0.805). The species *A. aspersa* inhabits Funka Bay in Japan, characterized by relatively cold waters with temperatures of 4–21˚C during the reproductive period [17]. This suggests that a colder environment might be conducive to optimizing egg transparency in this species in Japan. *Phallusia philippinensis* in subtropical seas and typically incubated at 20–25˚C [18], displays eggs with remarkably high transparency [14, 19]. *A. aspersa* and *P. philippinensis* belong to the same family, Ascidiidae. Investigating the mechanisms that optimize egg transparency in these two closely related species within diverse habitats marked by different temperatures presents an intriguing avenue for further research.

### Relationships between salinity and bio-transparency

Salinity positively correlates with bio-transparency across the 0 to 41 ppt range (Fig 1, $R^2$ = 0.929). This range encompasses the salinity tolerance limits for *A. aspersa*, which spans from 18 to 40 ppt [20], while notably excluding the extreme condition of 55 ppt. *A. aspersa* residing in Funka Bay encounters a relatively consistent salinity level ranging from 31–34 ppt [17]. Interestingly, our findings diverge from conventional expectations, as elevated salinity tends to render grass shrimp opaque. This phenomenon arises due to an uneven reflective index resulting from accumulation of excess fluid in the extracellular space between muscle fibers [4]. This discrepancy could signify distinct responses in single-cell and multicellular organisms concerning shifts in organismal transparency.

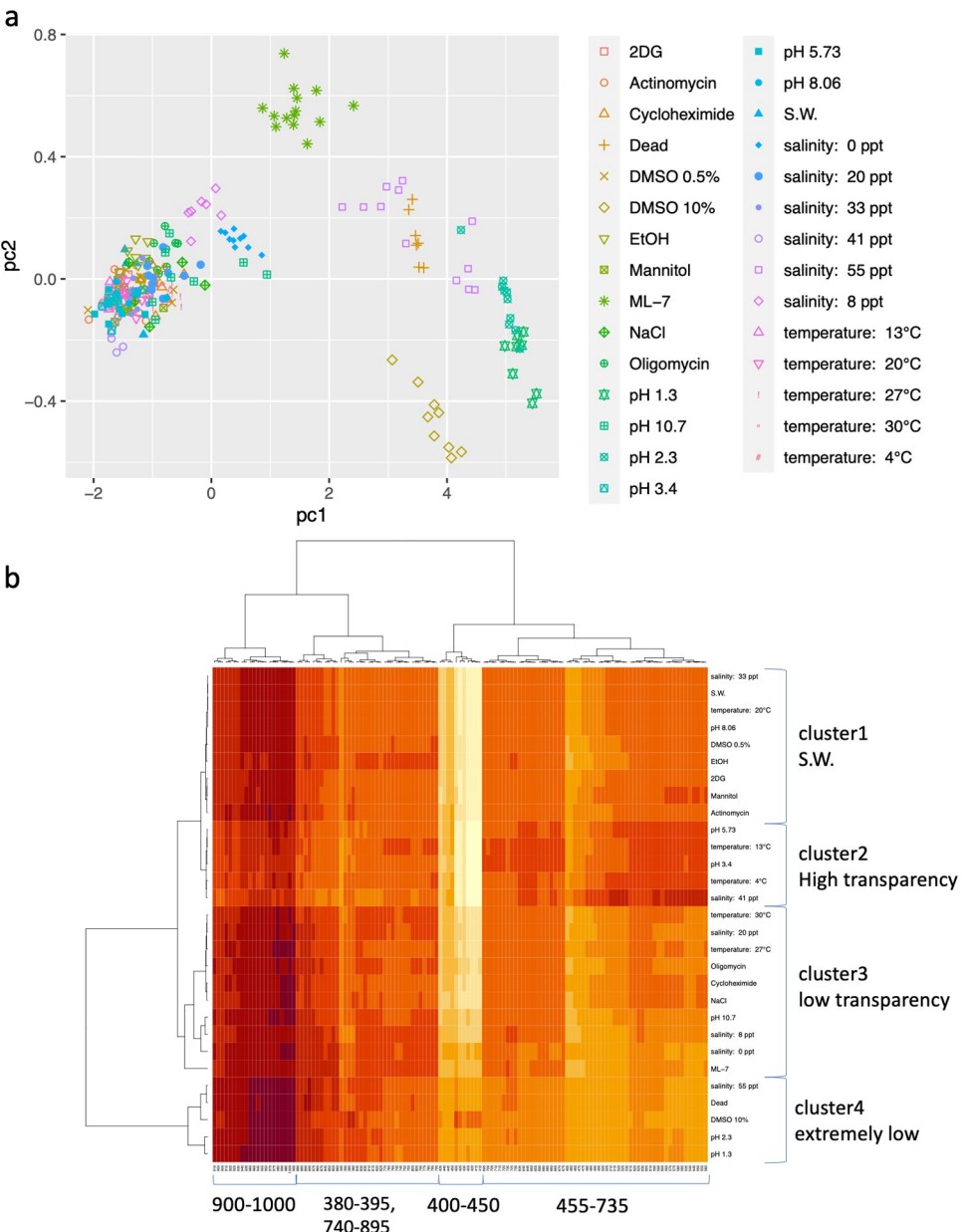

**Fig 3. PCA map and heatmap. a** PCA plot of *A. aspersa* eggs' transparency data (n = 272). Dimension reduction was performed to 125 band transmittances of each ascidian egg. PC1 and PC2 explained 98.9% and 0.79% of the total variance, respectively. **b** Heatmap of the same ascidian eggs' transparency data.

## Relationships between the pH and bio-transparency

Extremely low pH levels (pH 1.3, 2.3) induced a transformation in the coloration of eggs, resulting in a significant increase in attenuation, with a peak attenuation point observed at 440 nm (S3C Fig). The ascidian *Halocynthia roretzi* eggs also exhibit a 440-nm peak, believed to stem from carotenoid xanthophylls [21]. Similarly, *Ciona robusta* show variety of carotenoids within the yolk of its eggs [22, 23]. The color alteration observed in *A. aspersa* eggs could potentially arise from activating metabolic pathways involving substances like carotenoids, as seen in biochemical reactions governing pigmented embryos in crustacean species [24]. *A.*

*aspersa* eggs might be repressed under balanced pH conditions, while carotenoids are recognized for their photoprotective and anti-oxidative roles in the embryos of marine invertebrate [25–27]. On the other hands, deviations towards low pH levels and raising temperature are likely to impact the three-dimensional structures of proteins accountable for transparency, such as alpha-crystallin—a major component of vertebrate lenses [28]. These variations possibly influence the three-dimensional configuration of some proteins, potentially serving as a modulator of transparency of ascidian eggs.

## Drugs affecting egg transparency

We conducted quantitative evaluation of transparency alterations by subjecting ascidian eggs to 11 drugs containing essential physiological inhibitors that affect various egg functions. Bio-transparencies were significantly changed by six drugs and diverged from 89.4% to 29.5%. Elevating osmotic pressure through high concentration of mannitol (250 mM) did not lead to any significant decrease in transparency. Particularly noteworthy was the substantial decrease in transparency caused by DMSO (Fig 2A). This effect could potentially be attributed to surface structure damage, leading to a disruption in the ordered reflective index of eggs. This disruption might result from high concentrations of DMSO can perturb membrane lipid structures and interfere with dehydration process [29].

Furthermore, the use of multiple inhibitors was associated with a wide array of cellular processes, encompassing metabolism (oligomycin-A), cytoskeletal structure (ML-7), transcription or translation (actinomycin and cycloheximide), and hydration (DMSO). Such inhibitors can influence various aspects of the organism's cellular functions. These findings collectively underscore the intricate relationship between diverse cellular processes and the transparency exhibited by ascidian eggs.

## Hyperspectral imaging and PCA analysis dissected of the mechanism of transparency

Transparency was drastically diminished under several conditions, including pH levels of 1.3 and 2.3, 10% DMSO, salinity of 55 ppt, dead eggs, and the presence of ML-7. Nevertheless, PCA effectively distinguished and positioned these factors differently. This differentiation implies that these factors elicit decreases in transmittance across distinct wavelength ranges. The emergence of these distinct groups like mirrors the diverse mechanisms underlying the reduction in transparency.

The bio-transparency was similar for 55 ppt salinity and 10% DMSO; however, they were differentiated by the PCA analysis. Specifically, PC2 exhibited bifurcation into two distinct clusters: one with higher values corresponding to 55 ppt salinity and another with lower values corresponding to 10% DMSO (Fig 3A). Within the PCA analysis, dead eggs were in a similar zone to the 55 ppt salinity. This alignment is hypothesized to stem from the rupturing of egg membrane due to osmotic pressure. In contrast, other contributing factors did not aligned closely with the distribution of dead eggs. This discrepancy suggest that these factors exerted a variable impact on the transparency of each egg along a spectrum, differing from the uniformity observed in cell death.

Utilizing PCA and clustering analysis in conjunction with hyperspectral imaging, it became evident that diverse cellular processes could underlie the transparency of ascidian eggs. Moreover, these processes might influence distinct segments of spectrums. This phenomenon is exemplified by the spectrum exhibiting a sharp decline at 415 nm due to the myosin light chain kinase inhibitor ML-7, in contrast to the gradual gradient reduction caused by 10% DMSO at the peak of 490 nm (Fig 2A). The feasibility of this analysis hinged on the

comprehensive wavelength measurements enabled by hyperspectral imaging. Our evaluation system is a powerful tool for dissecting the complexity of bio-transparency mechanisms and holds promise for its applicability to other organisms. A comparative analysis across species could potentially yield further enlightening insight.

## Conclusion

By combining hyperspectral imaging across a broad wavelength spectrum with PCA analysis, we established an evaluation system capable of quantifying transparency alterations in micro-level organisms. This framework integrates a range of multi-modal factors encompassing external environmental variables and internal molecular constituents. Our initial application focused on the examination of transparent ascidian eggs. The shifts in transparency within these eggs were quantitatively gauged and clarified across varying wavelength tiers. Notably, bio-transparency exhibited positive correlation with temperature but a negative correlation with salinity. Furthermore, among 11 drugs targeting diverse cellular processes, seven exhibited significant decrease in bio-transparency. Utilizing PCA, we categorized the factors responsible for transparency changes, revealing diverse mechanisms through which opacification influenced distinct spectral regions. Our evaluation system holds the potential to enhance comprehension of organismal transparency mechanisms and how a convergence of multi-modal factors modulate them.

## Supporting information

**S1 Table. Calculated bio-transparencies and relative attenuation coefficients of all experiments.**
(PDF)

**S2 Table. Egg diameters and relative expansion rates calculated as the ratio of diameter between each condition and control condition (20˚C, 33 ppt, pH 8.06).**
(PDF)

**S1 Fig. Stereomicroscopy pictures of ascidian eggs.**
(PDF)

**S2 Fig. Transmittance spectra of ascidian eggs.**
(PDF)

**S3 Fig. Relative attenuation coefficients when changing environmental factors.**
(PDF)

**S4 Fig. Relative attenuation coefficients when drug treatments.**
(PDF)

**S5 Fig. Attenuation spectra of ascidian eggs.**
(PDF)

**S6 Fig. Relative attenuation coefficients of clustering.**
(PDF)

**S1 File. Normalized transmittance values of all experiments.**
(XLSX)

## Acknowledgments

We thank Dr. Minoru Ikeda and Captain Toyokazu Hiratsuka (Onagawa Field Center of Tohoku University), Dr. Makoto Kanamori, Dr. Masafumi Natsuike, and Takuya Mizukami (Hokkaido Research Organization, Hakodate Fisheries Research Institute), Dr. Gaku Kumano (Graduate School of Life Sciences, Tohoku University), Mr. Akio Takiya, Dr. Tatsunari Mori, Dr. Takaaki Kayaba, and Dr. Motohito Yamaguchi (Hokkaido Research Organization, Central Fisheries Research Institute) for their help in valuable information and collecting ascidian samples. We also thank Dr. Noburu Sensui, Dr. Euichi Hirose, and Dr. Takeshi Onuma for their helpful comments. We thank Haruka Miyama Funakoshi for handling *A. aspersa*.

## Author Contributions

**Conceptualization:** Takumi T. Shito, Kohji Hotta.

**Data curation:** Takumi T. Shito.

**Formal analysis:** Takumi T. Shito.

**Funding acquisition:** Takumi T. Shito, Kohji Hotta.

**Investigation:** Takumi T. Shito.

**Methodology:** Takumi T. Shito, Kotaro Oka, Kohji Hotta.

**Project administration:** Kohji Hotta.

**Resources:** Takumi T. Shito, Kohji Hotta.

**Supervision:** Kotaro Oka, Kohji Hotta.

**Validation:** Takumi T. Shito.

**Visualization:** Takumi T. Shito.

**Writing – original draft:** Takumi T. Shito, Kohji Hotta.

**Writing – review & editing:** Takumi T. Shito, Kotaro Oka, Kohji Hotta.

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
