## [Decision Letter · Decision Letter 0]

15 Aug 2023

PONE-D-23-14380Multimodal factor evaluation system for organismal transparency by hyperspectral imagingPLOS ONE

Dear Dr. Hotta,

Thank you for submitting your manuscript to PLOS ONE. After careful consideration, we feel that it has merit but does not fully meet PLOS ONE’s publication criteria as it currently stands. Therefore, we invite you to submit a revised version of the manuscript that addresses the points raised during the review process.

We look forward to receiving your revised manuscript.

Kind regards,

Satheesh Sathianeson, Ph.D

Academic Editor

PLOS ONE

Journal Requirements:

2.Thank you for stating in your Funding Statement: 

"This study was supported in part by JSPS KAKENHI (21H00440, 23H04717), Keio University Research and Education Center for Natural Sciences Budget, and KLL Keio Leading Program to KH. The Research Institute of Marine Invertebrates (IKU2021-02) supported TTS. The Keio University Doctorate Student Grant-in-Aid Program from Ushioda Memorial Fund supported TTS. JSPS KAKENHI Grant Number JP 22J22628 supported TTS."

"This study was supported in part by JSPS KAKENHI (21H00440, 23H04717), Keio University Research and Education Center for Natural Sciences Budget, and KLL Keio Leading Program to KH. The Research Institute of Marine Invertebrates (IKU2021-02) supported TTS. The Keio University Doctorate Student Grant-in-Aid Program from Ushioda Memorial Fund supported TTS. JSPS KAKENHI Grant Number JP 22J22628 supported TTS."

Reviewers' comments:

Reviewer's Responses to Questions

**Comments to the Author**

1. Is the manuscript technically sound, and do the data support the conclusions?

Reviewer #1: Yes

Reviewer #2: Yes

2. Has the statistical analysis been performed appropriately and rigorously? 

Reviewer #1: Yes

Reviewer #2: Yes

3. Have the authors made all data underlying the findings in their manuscript fully available?

Reviewer #1: Yes

Reviewer #2: Yes

4. Is the manuscript presented in an intelligible fashion and written in standard English?

Reviewer #1: No

Reviewer #2: Yes

5. Review Comments to the Author

Reviewer #1: The manuscript 'Multimodal factor evaluation system for organismal transparency by hyperspectral imaging' by Shito et al. describes a method to evaluate the impacts of modulating media conditions and cell physiology on the transparency of the eggs of the ascidian Ascidiella aspersa. The core of the method (hyperspectral transparency measurement) is based on a previous report published in Scientific Reports (Shito et al., 2020) by the same group. Here, they show that various environmental parameters (temperature, salinity, pH and culture media) and pharmacological treatments do modify egg's transparency. Although the biological significance is very preliminary, the proof-of-concept of this approach is achieved.

Here are minor points I suggest to address in order to improve the quality of the manuscript:

- careful proofreading should be performed since the manuscript is, at times, difficult to read. It seems for example that some words are missing in lines 51 and 76.

- line 83: explain how pH was adjusted.

- the word 'inhibitors' seems inadequate for some of the treatments (mannitol, NaCl,...).

- lines 263-265: the pH of the body fluids or the tunic does not seem relevant regarding the egg's transparency.

- line 284: a potentially interesting outcome of the method and results is that different cellular processes may be involved and that they could impact different regions of the spectrum. The authors could help the reader by highlighting some clear examples.

- the authors do not comment on the fact that transparency in normal conditions seems maximal/optimal since none of the conditions robustly increases transparency. Have the authors tried to combine the different conditions that increase transparency (low temperature, high salinity and 2DG)?

- the authors take into consideration variation/changes in egg diameter by calculating an 'expansion rate'. However, it is not clear whether this rate is used to adjust the transmittance values that are displayed on the figures. I guess the 'attenuation coefficient' does take this parameter into consideration, but it was overall unclear.

- the measures were made after 21hrs incubation. Could the authors comment on the speed of transparency changes?

Reviewer #2: The munscript by Shito et al. presented the establishment of the method for evaluation of organsmal tranparency and its utility. The authors demontrated the quantification of the effects of pH, temparatures,salinity, and several inhibitors transparency of ascidian (Ascidiella aspersa) eggs. In particular, they verified that the trasparency mechanisms varied among these environmental or phamacological factors. The methods are appropriate, the results are clear, and the discussion is sufficient. Overall, this manucript is well-written, and the presented method will contribute to advances in investigations of organimal transparency. Consequently, the reviewer makes a few minor comments as below.

(1) The reviewer wonders whether the ascidian eggs were all vital under experimental conditions. For example, actynomycin D (a transcriptional inhibitor) and cycloheximide (translational inhibitor) frequently causes apoprosis or other cell death. Moreover, such low pH may be harmful to the eggs, while the tunic of the adult ascidians are resistant against it. Otherwise, The reviewer also speculates that the viability of the eggs could be ignored in this research. Please state abot it.

(2) In the Discussion. The authors discuss about carotenoids as pH-sensitive compounds responsible for transparency. The reviewer guesses that Low anf high pH are expected to affect 3-dimentional structures of proteins responsible for transparency (e.g., alpha-crystallin that is a major component of mammalian lens) . Likewise, tmepaerature also affects 3-dimentional structures of proteins, which may also modulate transparency. If possible, please discuss transparency in light of 3-demenotinal structural changes induced by alteration of pH and temperature.

(3) Line 30. "suggesting" is better than "which suggested"; Line 240. Insert a space after "ascidian".

6. PLOS authors have the option to publish the peer review history of their article (what does this mean?). If published, this will include your full peer review and any attached files.

Reviewer #1: No

Reviewer #2: **Yes: **Honoo Satake

---

## [Author Response · Author response to Decision Letter 0]

7 Sep 2023

We thank the reviewers for the important comments and suggestions to our manuscript. We included all of them in this new version; we think that they greatly improved it.

Hoping that the manuscript can be considered for the publication in PLoS ONE.

---

## [Decision Letter · Decision Letter 1]

25 Sep 2023

Multimodal factor evaluation system for organismal transparency by hyperspectral imaging

PONE-D-23-14380R1

Dear Dr. Hotta,

We’re pleased to inform you that your manuscript has been judged scientifically suitable for publication and will be formally accepted for publication once it meets all outstanding technical requirements.

Kind regards,

Satheesh Sathianeson, Ph.D

Academic Editor

PLOS ONE

Additional Editor Comments (optional):

Reviewers' comments:

Reviewer's Responses to Questions

**Comments to the Author**

1. If the authors have adequately addressed your comments raised in a previous round of review and you feel that this manuscript is now acceptable for publication, you may indicate that here to bypass the “Comments to the Author” section, enter your conflict of interest statement in the “Confidential to Editor” section, and submit your "Accept" recommendation.

Reviewer #1: All comments have been addressed

Reviewer #2: All comments have been addressed

2. Is the manuscript technically sound, and do the data support the conclusions?

Reviewer #1: Yes

Reviewer #2: Yes

3. Has the statistical analysis been performed appropriately and rigorously? 

Reviewer #1: Yes

Reviewer #2: Yes

4. Have the authors made all data underlying the findings in their manuscript fully available?

Reviewer #1: Yes

Reviewer #2: Yes

5. Is the manuscript presented in an intelligible fashion and written in standard English?

Reviewer #1: Yes

Reviewer #2: Yes

6. Review Comments to the Author

Reviewer #1: (No Response)

Reviewer #2: The authors have addressed all issues raised by the reviewer. Cosequently, the reviwer recommends acceptance for publication in PLOS One.

7. PLOS authors have the option to publish the peer review history of their article (what does this mean?). If published, this will include your full peer review and any attached files.

Reviewer #1: No

Reviewer #2: **Yes: **Honoo Satake

---

## [Editor Report · Acceptance letter]

28 Sep 2023

PONE-D-23-14380R1 

Multimodal factor evaluation system for organismal transparency by hyperspectral imaging 

Dear Dr. Hotta:

I'm pleased to inform you that your manuscript has been deemed suitable for publication in PLOS ONE. Congratulations! Your manuscript is now with our production department. 

Kind regards, 

on behalf of

Dr. Satheesh Sathianeson 

Academic Editor

PLOS ONE